



# Simulating barrier island response to sea-level rise with the barrier island and inlet environment (BRIE) model v1.0

Jaap H. Nienhuis[1*] and Jorge Lorenzo-Trueba[2]

[1] Department of Physical Geography, Utrecht University, Utrecht, NL
[2] Earth and Environmental Studies, Montclair State University, Montclair, New Jersey, USA

*Correspondence to*: Jaap H. Nienhuis (j.h.nienhuis@uu.nl)

The authors declare that they have no conflict of interest.

**Abstract.** Barrier islands are low-lying coastal landforms vulnerable to inundation and erosion by sea-level rise. Despite their socio-economic and ecological importance, their morphodynamic response to sea-level rise or other hazards is poorly understood. To tackle this knowledge gap, we outline and describe the BarrieR Inlet Environment (BRIE) model that can simulate long-term barrier morphodynamics. In addition to existing overwash and shoreface formulations, BRIE accounts for alongshore sediment transport, inlet dynamics, and flood-tidal delta deposition along barrier islands. Inlets within BRIE can open, close, migrate, merge with other inlets, and build flood-tidal delta deposits. Long-term simulations reveal complex emergent behaviour of tidal inlets resulting from interactions with sea-level rise, and overwash. BRIE also includes a stratigraphic module, which demonstrates that barrier dynamics under constant sea-level rise rates can result in stratigraphic profiles composed of inlet fill, flood-tidal delta and overwash deposits. In general, the BRIE model represents a process-based exploratory view of barrier island morphodynamics that can be used to investigate long-term risks of flooding and erosion in barrier environments. For example, BRIE can simulate barrier island drowning in cases where the imposed sea-level rise rate is faster than the morphodynamic response of the barrier island.

## 1 Introduction

Barrier islands are long, narrow, sandy stretches of land that occupy a significant fraction of modern coastlines around the world. Barriers are often densely populated, support diverse ecological communities, and protect bays and wetlands that provide a range of ecosystem services (McLachlan 1983, Barbier et al. 2011). Despite their economic and ecological importance, there exists a critical gap in understanding how barriers respond to coastal change generally, and sea-level rise (SLR) specifically. A necessary condition for barrier islands to migrate landwards and keep up with SLR is that sufficient sediment transport from the barrier front to the top and back via overwash fan deposition and flood-tidal delta formation (Kraft, 1971; Armon & McCann, 1979; Inman & Dolan, 1989; Mallinson et al., 2010; Moore et al., 2010; Lorenzo-Trueba & Ashton, 2014 or LTA14). There is little information, however, regarding the relative roles of overwash and tidal fluxes in determining the rate of barrier landward migration, and the ability of barriers to keep pace with SLR.



Here we present the BarrieR Inlet Environment (BRIE) model to address this fundamental knowledge gap. Transgression in the model is driven by two main processes: overwash sedimentation and flood-tidal delta deposition (Leatherman, 1979; Pierce, 1969, 1970). To date, models aimed to assess barrier island change over geological timescales, typically account for only storm overwash, which is more suitable for a cross-sectional framework. Tidal inlets, however, have been suggested to

5    contribute a large fraction of the transgressive sediment movement in a number of field studies (Pierce, 1969, 1970). The BRIE model extends the LTA14 formulation in the alongshore direction, and incorporates tidal inlet morphodynamics through Delft3D derived parameterizations (Nienhuis & Ashton 2016 or NA16). The purpose of the model is twofold, (i) to better understand long-term barrier island morphodynamics, including effects of, for example, sea-level rise, human development (jetties, beachs nourishment), or storm pattern changes, and (ii) to improve paleo environment reconstructions.

Section 2 of this manuscript provides a background on barrier island environments and recent model developments. In section 3, we discuss model formulations, including overwash fluxes, alongshore sediment transport, and tidal inlet morphodynamics. Section 4 includes a model run that demonstrates the capabilities of the BRIE framework, including inlet dynamics alongshore, and the generation of alongshore stratigraphic profiles. Section 5 explores model sensitivity to grid

and time resolution, as well as a comparison to other barrier island models. We conclude with a few exploratory results and a discussion of potential model applications.

## 2 Background

### 2.1 Barrier islands and SLR

Barrier islands are narrow strips of land, formed by waves through a variety of (hypothesized) mechanisms (e.g., Gilbert,

1885; McGee, 1890; Penland et al., 1985), associated with relatively slow SLR rates and primarily passive margins (FitzGerald et al. 2008, Stutz and Pilkey 2011, McBride et al. 2013). The emergence of many barrier islands can be traced back to about 6,000 years before present, when Holocene SLR slowed down (McBride et al., 2013).

However, the relationship between barrier islands and SLR is complex. Under no SLR, barrier islands are generally not

observed as their associated back-barrier environments would fill completely (e.g., Beets and van der Spek, 2000). In contrast, under moderate SLR rates marshes and tidal flats generally occupy backbarrier environments. In this case, in order to maintain their elevation respect to sea level, barriers migrate towards land as storm overwash and flood tidal flows deposit sediment. Under higher SLR rates, however, it is more difficult for barriers to maintain their subaerial portion above sea level. Consequently, when onshore-directed sediment fluxes are insufficient, barrier islands drown in place and are left

offshore (Rodriguez et al. 2001, Mellet 2012). Additionally, when onshore-directed sediment flux events are very intense and frequent, barrier islands are unable to maintain their geometry as they rapidly migrate towards land, which also results in drowning (LTA14). This potentially delicate balance between SLR and barrier response, together with the current





projections of future acceleration in SLR, highlight the need to better constrain onshore-directed sediment fluxes in different barriers island systems (Carruthers et al., 2013; Lazarus, 2016; e.g., McCall et al., 2010; Rogers et al., 2015).

### 2.2 Barrier overwash

One way for sediment to be transported across the barrier is through storm overwash. Differences in water level setup
between the ocean and the lagoon during a storm can force the flow of water and sediment through and above the subaerial portion of the barrier. Most frequently this flow is directed landward, resulting in transport of sediment from the ocean to the bay side where it deposits as the flow spreads laterally into the lagoon (Carruthers et al., 2013; Donnelly et al., 2006). Although this process is complex and highly intermittent, individual storm events integrated over time result in a net landward sediment flux, which allows barriers to keep pace with SLR over geological time scales (Leatherman, 1983).
Despite its importance in terms of future barrier island morphodynamic response and vulnerability to flooding (Miselis and Lorenzo-Trueba, 2017), this long-term landward sediment flux is generally poorly constrained, and its relationship with modern overwash fluxes largely unexplored (Carruthers et al., 2013; Donnelly et al., 2006; Lazarus, 2016; Rogers et al., 2015). This lack of constraints on long-term overwash fluxes has resulted in a suit of barrier island models that do not compute overwash processes as a function of single storm events. Instead, such models parameterize overwash volume
fluxes as a function of barrier geometry and observations of barrier island migration. For example, Leatherman (1979) observed that narrow barrier islands tend to be more susceptible to overwash events than wide barrier islands. He defined a 'critical barrier width' below which overwash is frequent and the barrier migrates rapidly, and above which overwash and barrier migration tend to be slow. Based on these findings, overwash is often parameterized by assuming the volume is inversely proportional to island width (e.g., Jiménez and Sánchez-Arcilla, 2004; LTA14), and additionally adjusted based on
local factors such as land use (e.g., Rogers et al., 2015).

### 2.3 Tidal Inlets

Aside from storm overwash, tidal inlets have also been found to be a major contributor to barrier transgression (Inman and Dolan, 1989; Moslow and Heron, 1978; Pierce, 1969). Tidal inlets derive their transgressive potential through the deposition of flood-tidal deltas. The volume of flood-tidal delta deposits correlates with the size of the associated inlet (Powell et al.,
2006). Simple equilibrium models (e.g., Stive et al., 1998) suggest that initially flood-tidal deltas grow fast but that their growth slows down as they approach an equilibrium volume and the bay fills up near the inlet. Inlet migration can therefore add to transgressive transport by exposing new bay to flood-tidal delta deposition (NA16). For these two reasons, it has been hypothesized that short-lived and rapidly migrating inlets are most efficient for barrier transgression (Pierce, 1970). However, if migration rate and life span of tidal inlets correlate with sediment import, their potential for transgression should
then depend on factors such as basin size, ocean waves, and tidal conditions.

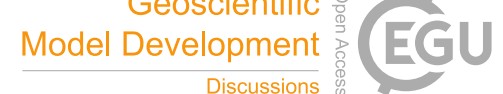

Along extended barrier coastlines, barrier morphodynamics is complicated by the existence of multiple tidal inlets. Tidal inlets interact through their control on water surface elevation in the tidal basin. This interaction can cause inlets to close or change size (van de Kreeke et al., 2008; Roos et al., 2013). Observations of tidal inlet spacing (Davis and Hayes, 1984), corroborated by a recent modeling study (Roos et al., 2013), found that increasing tidal range and basin size can allow inlets

to exist closer together. For a barrier coast, a greater number of inlets likely enhances their contribution to barrier transgression.

## 2.3 Previous Numerical Modeling Efforts

The joint long-term effect of storm overwash and tidal inlets on barrier island evolution remains difficult to quantify. On the one hand, engineering models typically assess barrier island changes over annual to decadal timescales, which includes

overwash fluxes and tidal inlet formation during storm events. For example, models such as XBeach (McCall et al., 2010; Roelvink et al., 2009) resolve wave dynamics coupled with sediment transport during storm events, and are able to capture barrier morphological changes, including breaching. On decadal timescales, models like Delft3D (Deltares, 2014) have been applied to study inlets but these typically do not include the effect of storms or SLR (e.g., Tung et al., 2009; NA16). On longer timescales, models no longer use laboratory-validated sediment transport relationships but rather use various degrees

of conceptual relationships between barrier geometry and barrier island movement (Cowell et al. 1995; Storms et al., 2002; Stolper 2005; Masetti et al., 2008; Wolinsky & Murray 2009; LTA 2014). Some of these models are morphokinematic; based upon the conservation of mass and maintenance of barrier geometry (Cowell et al. 1995, Wolinsky & Murray 2009, Stolper 2005). The models developed by Storms et al. (2002), Masetti et al. (2008), and LTA14, are morphodynamic as they account for sediment fluxes along the shoreface and across the barrier island. LTA14 represents a significant simplification

compared to other morphodynamic models, making it suitable for model extensions and model coupling (such as the one presented here).

Coming from a different angle, ASMITA has been developed in part to understand the effects of SLR on inlets and their back-barrier environments (Van Goor et al., 2003; Stive et al., 1998; Townend et al., 2016). ASMITA couples coasts to their

back-barrier environment via sediment exchanges determined by the deviation of a morphological element (ebb delta, tidal flat, etc.) from an assumed equilibrium volume. Inlets cannot close or migrate, and the model does not account for overwash processes (Stive et al., 1998). In ASMITA, as well as other back-barrier models (Van Maanen et al., 2013; Mariotti and Canestrelli, 2017), the maximum potential sediment import through tidal inlets exerts a first-order control on the ability of back-barrier environments to sustain themselves during SLR.

Here, we describe a new model (BRIE) that accounts for tidal inlet dynamics, including opening, closing and lateral migration, combined with barrier overwash processes as described by LTA14. Within the realm of coastal geomorphological models, BRIE can be considered a "Large-Scale Coastal Behavioral (LSCB)" model (de Vriend et al., 1993). It seeks to

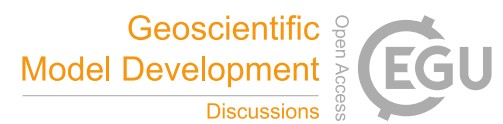

represent only the main governing mechanisms of the coast at appropriate time and scale scales, without fully resolving the mechanics of fluid and sediment transport. There is a rich body of literature concerning LSCB models, ranging from rocky coasts (Walkden and Hall, 2011), barrier islands (Stolper et al., 2005; LTA14), tidal basins (Townend et al., 2016), tidal inlets (Kraus, 2000), sandy coastlines (Ashton et al., 2001), as well as aggregates of LSCB models that couple these elements (Ashton et al., 2013; Payo et al., 2017). To our knowledge, BRIE would be the first to explicitly couple barrier islands and tidal inlet morphodynamics. Despite its simplicity, BRIE provides a novel approach to study the evolution of barrier islands under decadal to millennial timescales. Moreover, it allows us to explore complex barrier dynamics across a wide range of parameter values.

## 3 Model

We developed the BRIE modeling framework to study barrier island response to SLR. The model incorporates longshore interactions by linking the cross-shore barrier island model presented by LTA14 in a series of dynamic cross-shore profiles. We apply the storm overwash and shoreface response functions independently in each cell (Fig. 1). Feedbacks between overwash dynamics alongshore arise through the coupling with alongshore sediment transport, which can adjust the shoreline location and influence the shoreface slope and barrier overwash. The model also accounts for the formation, closing and migration of tidal inlets following the parameterizations from NA16 (Fig. 1).

To our knowledge, this is the first morphodynamic model for long-term (decadal to millennial time scales) barrier island evolution that accounts for both tidal and overwash sediment fluxes. The model is written in MATLAB, and a typical runtime for a 100 km long barrier island stretch over a 10,000 year simulation is ~1 minute. Its simplicity and computational speed enables us to explore model behavior for a wide range of parameter values.

### 3.1 General description and model setup

After initializing the environment (typically ~100 km long barrier island with periodic boundaries) and determining wave climate and shoreface parameters (Table 1), we run each time step in a for-loop. In each iteration, we first raise sea level (Fig. 2). SLR affects subaerial barrier volume and shoreface slope, which in turn drives overwash and shoreface fluxes (Section 3.2).

Next, we determine if new inlets should be formed (Sect. 3.3.1), in which case we analyze their hydrodynamics and calculate their equilibrium dimensions. For each inlet, we distribute sediments into the flood-tidal delta, the barrier island, and the shoreface (Section 3.3.3). Flood-tidal delta deposition changes the back-barrier location (Fig. 1). After each timestep, we add the different sources and sinks to the coastal zone including diffusive wave-driven alongshore sediment transport, and





implicitly determine a new shoreline, back-barrier, and shoreface toe position. See table 1 for an overview of all model parameters and units.

### 3.2 Cross-shore morphodynamics

### 3.2.1 Cross-shore barrier model

At a minimum, barrier islands can be described in the cross-shore dimension as composites of three regions: the active shoreface on the ocean side, the subaerial portion of the barrier island, and the back-barrier lagoon on the terrestrial side, where infrequent overwash processes determine the volume of onshore-directed sediment fluxes (LTA14). Note that the overwash model is applied independently for every alongshore cell $j$, from 1 to $n_y$ (Fig. 1), but we leave out these indices for clarity. Assuming an idealized geometry, the cross-shore evolution of the barrier system can be fully determined with the

rates of migration of the shoreface toe,

$$\frac{\Delta x_t}{\Delta t} = 4Q_{sf} \frac{H + D_T}{D_T (2H + D_T)} + \frac{2\dot{z}}{s_{sf}} \, , \tag{1}$$

the shoreline,

$$\frac{\Delta x_{s,ow,sf}}{\Delta t} = \frac{2(Q_{ow,b} + Q_{ow,h})}{(2H + D_T)(1 - f)} - 4Q_{sf} \frac{H + D_T}{(2H + D_T)^2} \, , \tag{2}$$

the backbarrier shoreline,

$$\frac{\Delta x_{b,ow}}{\Delta t} = \frac{Q_{ow,b}}{H + D_{lagoon}} \, , \tag{3}$$

and the barrier height above sea level,

$$\frac{\Delta H_{ow}}{\Delta t} = \frac{Q_{ow,h}}{W_b} - \dot{z} \, . \tag{4}$$

where $Q_{sf}$ is the sediment flux at the shoreface, $\dot{z}$ is the SLR rate, $Q_{ow,h}$ is the top-barrier overwash component, and $Q_{ow,b}$ is the back-barrier overwash component. Other variables and parameters are defined in table 1. Note that equations 1-4 follow

the barrier island model of LTA14 except for the $(1-f)$ factor in eq. 2 that accounts for fine-grained sediment in the back-barrier. We discuss this modification in section 3.2.2.

  We compute overwash flux using a simple formulation that assumes the existence of a critical barrier width $W_{b,crit}$ and a critical barrier height $H_{crit}$ beyond which there is no overwash to the back and the top of the barrier, respectively. When the

barrier width $W_b$ and height $H$ are below their critical values, the overwash rates $Q_{ow,h}$ and $Q_{ow,b}$ scale with their associated deficit volumes, $V_{d,h}$ and $V_{d,b}$, resulting in a overwash flux heightening the barrier,


$$Q_{ow,h} = Q_{ow,max} \frac{V_{d,h}}{\max(V_{d,b} + V_{d,h}, H_{crit} \cdot W_{b,crit})} , \quad (5)$$

and an overwash flux widening the barrier,

$$Q_{ow,b} = Q_{ow,max} \frac{V_{d,b}}{\max(V_{d,b} + V_{d,h}, H_{crit} \cdot W_{b,crit})} , \quad (6)$$

We define the volume deficits with respect to an equilibrium defined by the critical barrier width and height (LTA14). In this
way, we can compute $V_{d,b}$ and $V_{b,h}$ as follows:

$$V_{d,b} = \max\left[0, \left(W_{b,crit} - W_b\right)\left(H + D_{lagoon}\right)\right] , \quad (7)$$

$$V_{d,h} = \max\left[0, \left(H_{crit} - H\right)W_b\right] . \quad (8)$$

The shoreface flux is controlled by the shoreface response rate $k_{sf}$ and the deviation of the shoreface slope from its
equilibrium slope,

$$Q_{sf} = k_{sf}\left(s_{sf,eq} - s_{sf}\right). \quad (9)$$

### 3.2.2 Modifications to the LTA14 barrier model

All the above formulations are identical to LTA14 except for eq. 2, which we adjust to account for fine sediments in the
backbarrier. LTA14 assumes a backbarrier depth geometrically determined as $z$-$x_b$·$s_{background}$ (Fig. 1), where $s_{background}$ is the
basement slope. This depth assumes the absence of backbarrier sediment deposition (i.e., $f=0$, where $f$ is the fine sediment
fraction), and therefore represents the upper limit depth. The BRIE model accounts for fine sediment deposition by selecting
a backbarrier depth $D_{lagoon}$ (see eq. 3) that is within the range $0 \leq D_{lagoon} \leq z$-$x_b$·$s_{background}$. We then compute the fine
sediment thickness in the backbarrier (Fig. 1) as:

$$D_{fines} = z - x_b s_{background} - D_{lagoon} . \quad (10)$$

In turn, we can geometrically define $f$ as follows:

$$f = \frac{D_{fines}}{D_{fines} + D_{lagoon} + H} , \quad (11)$$

As barriers migrate towards land, fine sediments are absorbed in the bay side and exported at the shoreface on the ocean
side; a dynamic that can play a significant role on the total barrier sediment volume changes (Brenner et al. 2015). BRIE
accounts for fine sediment export at the shoreface by assuming that the fine sediment fraction $f$ given by equation (11) is
representative of the entire cross-section of the barrier, and that the sediment exchange between the upper and lower
shoreface $Q_{sf}$ is not affected by the presence of fine sediments. In this way, eq. (2) accounts for the fact that the fine sediment
fraction $f$ of the overwash sediment volume extracted from the shoreface does not contribute to the total volume of the



barrier. In other words, only a fraction (*1-f*) of the shoreface volume eroded (i.e., $\Delta x_{s,ow,sf}(2H + D_T)$) deposits on top and/or back of the barrier (Fig. 3).

### 3.2.3 Parameter estimation for the LTA14 model

The cross-shore barrier model is a function of a number of parameters, including the shoreface depth $D_T$, the equilibrium shoreface slope $s_{sf,eq}$, and the shoreface response rate $k_{sf}$. These three parameters, although generally poorly constrained, can be estimated as a function of wave and sediment characteristics (e.g., characteristic sediment grain size $D_{50}$, significant wave height $H_s$). This allows us to investigate how storm overwash, alongshore transport, and inlet dynamics co-vary for a particular environment.

The shoreface response rate can be viewed as the integrated cross-shore sediment transport flux between a depth $z_0$ below wave breaking, and the shoreface depth $D_T$ (LTA14; Ortiz and Ashton, 2016). Here we integrate the shoreface flux $k_{sf}$ (converted from m/s into units of m/yr),

$$k_{sf} = (3600 \cdot 24 \cdot 365) \frac{\int_{z_0}^{D_T} \phi \frac{H(z)^5 g^{5/2}}{32 w_s^2 z^{5/2}} dz}{D_T - z_0}, \tag{12}$$

where $\phi = 16 e_s C_s \rho /(15\pi(\rho_s - \rho)g)$ and $H(z)$ is the local wave height at depth $z$. We solve this integral assuming $H(z)$ is a shallow water wave that can be estimated by the offshore wave climate and a shoaling coefficient, $H(z) = H_s \sqrt{\sqrt{g} \cdot T / 4\pi / \sqrt{z}}$. We derive a simple analytical expression of the integrated shoreface response rate,

$$k_{sf} = (3600 \cdot 24 \cdot 365) \frac{e_s c_s g^{\frac{11}{4}} H_s^5 T_p^{\frac{5}{2}}}{960 R \pi^{\frac{7}{2}} w_s^2} \cdot \frac{\frac{1}{\frac{11}{4} z_0^{\frac{11}{4}}} - \frac{1}{\frac{11}{4} D_T^{\frac{11}{4}}}}{D_T - z_0}, \tag{13}$$

where we estimate $z_0$ as the breaking wave depth $H_s/\gamma$, and $\gamma$ is 0.4 (Sallenger and Holman, 1985).

We determine the shoreface depth $D_T$ (m) using an empirical relationship based on the wave characteristics (Hallermeier, 1981),

$$D_T = 0.018 H_s T_p \sqrt{\frac{g}{R D_{50}}}. \tag{14}$$

We estimate the shoreface equilibrium slope $s_{sf,eq}$ as the slope at the depth of closure (Lorenzo-Trueba and Ashton, 2014),

$$s_{sf,eq} = \frac{3 w_s}{4\sqrt{g D_T}} \left( 5 + \frac{3 T_p^2 g}{4\pi^2 D_T} \right), \tag{15}$$

where the settling velocity is calculated based on the empirical formulation developed by Ferguson and Church (2004),



$$w_s = \frac{RgD_{50}^2}{18 \cdot 10^{-6} + \sqrt{\frac{3}{4} RgD_{50}^3}} \ .$$ (16)

### 3.3 Inlet model

Inlets can form along barrier island chains if there is sufficient potential for tidal flow between the lagoon and the open ocean (Escoffier, 1940). In turn, the potential for tidal flow is determined by factors influencing the potential tidal prism (e.g., the proximity of other tidal inlets nearby, the width and depth of the basin and the barrier, the marsh cover), and factors reducing tidal flow (e.g., tidal inlet friction, wave-driven transport into tidal inlets). Once inlets exist, they alter barrier morphodynamics by distributing sediments and enhancing storm overwash potential.

#### 3.3.1 Inlet formation

We allow the model to form new tidal inlets every $T_{storm}$ years at the location of minimum barrier volume $A_{barrier}$, where $T_{storm}$ can be considered as a storm return time. An inlet can only form at a distance of at least $L_{min}$ away from current inlets, where $L_{min}$ is a minimum inlet spacing (Roos et al., 2013). Although $L_{min}$ is likely dependent on a wide range of factors, we are not aware of field constrains on its value and therefore choose a constant $L_{min}$. We do not open a new inlet if the flow velocity through a new inlet is insufficient (see below). If a new inlet is opened, we place the barrier volume in the flood-tidal delta by increasing the backbarrier location,

$$\Delta x_{b,breach} = W_b \frac{H + D_{inlet}}{D_{lagoon}} \ .$$ (17)

with the implicit assumption that the flood-tidal delta top is approximately at sea level. Although inlets cannot open closer than $L_{min}$ away from existing inlets, differences in inlet migration rates can cause inlets to exist closer to each other (and merge). Additionally, inlets can also form when a section of the barrier drowns (negative barrier cross-sectional volume, i.e., $A_{barrier} < 0$), regardless of the distance to other existing inlets.

#### 3.3.2 Inlet hydrodynamics

At every timestep, we compute the distance among all inlets. Assuming the lagoon water drains to the nearest inlets, we determine the lagoon area per tidal inlet (the potential for tidal prism) by multiplying the water surface area (i.e., $W_{lagoon} \cdot L_{lagoon}$) with a predefined fraction occupied by marshes, $f_{marsh}$.

We compute inlet characteristics such as cross-sectional area and flow velocities based on de Swart and Zimmerman (2009), who in turn followed ideas established by Escoffier (1940) (Fig. 4). We solve the inlet area-velocity relationship (Escoffier, 1940; de Swart and Zimmerman, 2009) analytically for $u = u_e$, meaning that inlets adjust to maintain an equilibrium tidal

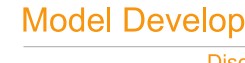 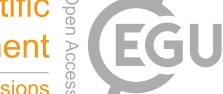

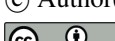

velocity amplitude where sediments will be neither deposited nor eroded. In this situation, the non-dimensionalized equilibrium inlet cross-sectional area is given by,

$$\tilde{A}_{inlet} = \frac{2\tilde{A}_H}{3} + \frac{2^{\frac{2}{3}} F_0}{6} + \frac{2^{\frac{1}{3}}(3 + \tilde{A}_H^2 \tilde{u}_e^2)}{3\tilde{u}_e^2 F_0}, \tag{18}$$

where $F_0$ is,

$$F_0 = \begin{pmatrix} 6\sqrt{3}\sqrt{\tilde{A}_H^3 \tilde{\gamma}^{-2} \tilde{u}_e^5 - \tilde{A}_H^4 \tilde{u}_e + 2\tilde{A}_H^2 \tilde{u}_e^{-1} - 9\tilde{A}_H \tilde{\gamma}^{-2} \tilde{u}_e^3 - 1 + \frac{27}{4} \tilde{\gamma}^{-4} \tilde{u}_e^7} \\ -27\tilde{\gamma}^{-2} \tilde{u}_e^2 - 2\tilde{A}_H^3 + 18\tilde{A}_H \tilde{u}_e^{-2} \end{pmatrix}^{\frac{1}{3}}. \tag{19}$$

In this formulation, $\tilde{A}_H = \omega W_b / \sqrt{g a_0}$ is a resonance non-dimensional cross-sectional area, $\tilde{u}_e = u_e / \sqrt{g a_0}$ is a non-dimensional equilibrium velocity, and $\tilde{\gamma}$ is the ratio of the potential tidal prism and the inlet friction,

$$\tilde{\gamma} = \frac{\gamma_{aspect} \sqrt{\omega L_{lagoon} W_{lagoon} (1 - f_{marsh}) \sqrt{a_0/g}}}{\frac{8}{3\pi} c_d W_b}, \tag{20}$$

where the a drag coefficient $c_d = g \cdot n^2 / D_{lagoon}^{\frac{1}{3}}$ (de Swart and Zimmerman, 2009).

Based on $\tilde{A}_{inlet}$ we determine the dimensional inlet cross-sectional area (in m$^2$),

$$A_{inlet} = \omega \cdot (1 - f_{marsh}) \cdot L_{lagoon} \cdot W_{lagoon} \cdot \sqrt{g a_0} \tilde{A}_{inlet}, \tag{21}$$

and the corresponding water velocity through the inlet,

$$u = \left( \frac{a_0}{2g} \tilde{\gamma} \tilde{A}_{inlet}^{\frac{1}{2}} \left[ -\xi + \left( \xi^2 + 4 \right)^{\frac{1}{2}} \right] \right)^{\frac{1}{2}}, \tag{22}$$

where $\xi = \tilde{\gamma} \tilde{A}_{inlet}^{\frac{1}{2}} \left( \tilde{A}_{inlet} - \tilde{A}_H \right)^2$. The inlet area function (eq. 21) evaluates to the largest cross-sectional area for which $u = u_e$ (1 m s$^{-1}$ in all simulations) if an equilibrium inlet area exists (Fig. 4). The inlet velocity evaluates to $u < u_e$ if an equilibrium does not exist (the friction through the inlet exceeds the potential tidal prism), at which point the inlet will close. Inlets adjust instantaneously to changes. Waves do not influence the size of the inlet, but alongshore sediment transport is assumed to be present to maintain an inlet to its equilibrium size.





### 3.3.3    Alongshore sediment transport into inlets

We calculate alongshore sediment transport into inlets ($Q_{s,in}$, converted from m³/s to m³/yr) based on the CERC formula recast into deep-water wave properties (AM06),

$$Q_{s,in} = (3600 \cdot 24 \cdot 365) \cdot k \cdot H_s^{12/5} \cdot T_p^{1/5} \cos\left(\varphi_0 - \theta\right)^{6/5} \sin\left(\varphi_0 - \theta\right), \tag{23}$$

where $k$ is a constant that is ~ 0.06 m³/⁵s⁻⁶/⁵ (Nienhuis et al., 2015), and $\phi_0$ is the wave direction. Shoreline orientation $\theta$ is defined by $\Delta x_s / \Delta y$. We determine the wave direction at every timestep from a cumulative distribution function defined by the wave asymmetry $a$ and wave highness $h$ (Ashton and Murray, 2006),

$$\phi_0 = f^{-1}(X), \tag{24}$$

$$f(\phi_0) = \begin{cases} a \cdot h \cdot \left(\frac{4}{\pi}x + 2\right) & \text{if } -\frac{1}{2}\pi < \phi_0 < -\frac{1}{4}\pi \\ a + \frac{4}{\pi}x(1-h)a & \text{if } -\frac{1}{4}\pi < \phi_0 < 0 \\ a + \frac{4}{\pi}x(1-h)(1-a) & \text{if } 0 < \phi_0 < \frac{1}{4}\pi \\ (1-a)h(\frac{4}{\pi}x - 2) + 1 & \text{if } \frac{1}{4}\pi < \phi_0 < \frac{1}{2}\pi \end{cases}, \tag{25}$$

where $x$ is uniformly distributed between 0 and 1.

Note that although we estimate sediment transport into inlets based on this method, we do not calculate shoreline change based on this particular wave angle at every time step. Instead, for model stability and efficiency, we calculate shoreline change using an implicit timestep non-linear diffusion equation, with inlets, storm overwash, and cross-shore shoreface transport acting as sediment sources or sinks (see section 3.5.2).

### 3.3.4    Inlet morphodynamics

After we have determined inlet cross-sectional area and wave-driven transport into the inlet, we distribute sediments between the updrift and downdrift portions of the inlet, and the flood tidal delta, following parameterizations of NA16 (Fig. 1). Inlets can migrate and erode into a barrier and also deposit a barrier. Inlets also form flood-tidal deltas. Ebb-tidal deltas are absent from this formulation because they do not present a sink from the littoral zone. Ebb-tidal deltas, however, implicitly determine the rate of inlet migration and the size of the flood-tidal delta through their effect on waves and currents (NA16). Inlet migration and flood-tidal delta deposition rates are dependent on the alongshore sediment transport into the inlet $Q_{s,in}$ and the sediment distribution fractions $\alpha$, $\beta$, $\delta$, $\beta_r$, $\alpha_r$, and $\delta_r$ (Fig. 1; NA16). These fractions are determined by Delft3D model experiments and parameterized as,

$$\alpha(I) = 1 - \beta(I) - \delta(I), \tag{26}$$

$$a_r(I) = 0.6 \cdot \alpha(I), \tag{27}$$





$$\beta(I) = \frac{1}{1+10\cdot I^3} \,, \tag{28}$$

$$\beta_r(I) = \frac{1}{1+0.9\cdot I^{-3}} \,, \tag{29}$$

$$\delta(I) = \frac{1}{1+3\cdot I^{-3}} \,, \tag{30}$$

$$\delta_r = \frac{A_{b,downdrift}\cdot\alpha - A_{b,updrift}\cdot\beta_r}{A_{b,updrift}} \,, \tag{31}$$

5  where $A_{b,updrift}$ and $A_{b,downdrift}$ are $W_b\cdot(D_{inlet}+H)$, the barrier cross-sectional area updrift and downdrift of the inlet, respectively.

We estimate the sediment distribution fractions based on the inlet momentum balance $I$, which is the ratio of the tidal and wave momentum flux $M_t$ and $M_w$,

$$I = \frac{M_t}{M_w}\cdot\frac{W_{inlet}}{W_b} = \frac{\rho_w u_{eq}^2 A_{inlet}}{\frac{1}{16}\rho_w g H_s^2 W_{inlet}}\cdot\frac{W_{inlet}}{W_b} \,. \tag{32}$$

For model stability, we depart from the original formulation of NA16 on two occasions:

(i) eq. 31 is a departure of the original formulation (NA16). The new function forces both inlet flanks to migrate at the same rate, making inlet width purely a function of inlet hydrodynamics.

(ii) we impose a maximum flood tidal delta volume following *Powell et al* (2006), such that $V_{fld,max} = 1\cdot10^4\left(u_e A_{eq}/2\omega_0\right)^{0.37}$ (m³). If this maximum is reached, we limit $I$ to 0.1 to ensure more efficient bypassing and inlet migration. In NA16, flood-tidal delta deposition is not a function of flood-tidal delta size, which in the case of this model would create unrealistically large flood-tidal deltas.

Based on the sediment distribution, the inlet can deposit sediment into the flood-tidal delta. Assuming that the flood tidal delta is at sea level, we can describe its rate of growth (Fig. 1c) as follows:

$$\frac{\Delta x_{b,inlet}}{\Delta t} = \frac{Q_{s,in}\left(\delta+\delta_r\right)}{W_{inlet}D_{lagoon}} \,, \tag{33}$$

change the sediment budget in the littoral zone,

$$\frac{\Delta x_{s,inlet}}{\Delta t} = \frac{\left(\beta+\beta_r\right)Q_{s,in}}{\left(H+D_T\right)W_{inlet}} \,, \tag{34}$$

and migrate alongshore in the direction of the littoral drift,





$$\frac{\Delta y_{inlet}}{\Delta t} = \frac{Q_{s,in}(\alpha + \alpha_r)}{A_{b,updrift}} . \tag{35}$$

Changes to the back-barrier and shoreline locations are estimated at every timestep. Inlet migration however, per time step $\Delta t$ (~ 0.05 yr), is typically much less than the alongshore discretization $\Delta y$ (~100 m). We therefore track inlet migration by assigning a 'fraction migrated' to one grid cell in each inlet. The inlet moves along the barrier if that fraction exceeds one or

drops below zero. New barrier island is constructed at sea level, $\Delta H_{inlet} = -H$. A second complication is that inlets are also typically (but not necessarily) wider than the alongshore discretization $\Delta y$. Inlets are therefore allowed to exist on multiple alongshore cells $j$, dependent on the inlet width, $n_{inlet,i} = W_{inlet} / \Delta y$, where $n_{inlet}$ is the number of alongshore cells that is taken up by inlet $i$ (Fig. 1).

### 3.4    Shoreline change

After we have determined the various sources and sinks of sediment to the nearshore environment, we distribute sediment alongshore between the different cells based on alongshore sediment transport. We use an implicit Crank-Nicolson scheme (Crank and Nicolson, 1947) to solve for shoreline change, governed by the following non-linear diffusion equation,

$$\frac{\Delta x_{s,j}}{\Delta t} = D_j \frac{\Delta^2 x}{\Delta y^2} + \frac{\Delta x_{s,j,ext}}{\Delta t} , \tag{36}$$

which includes the effect of wave refraction and shoaling and is therefore suitable to apply based on offshore (deep-water)

wave conditions (AM06). We have added a source/sink term $x_{s,j,ext}$ (m) to account for cross-shore sediment movement. $D_j$ is a non-linear term and accounts for the fact that diffusivity depends on the wave approach angle (AM06),

$$D(\theta) = \frac{k}{(H_b + D_T)} H_0^{12/5} T^{1/5} \left[ E(\phi_0) * \Psi(\phi_0 - \theta) \right], \tag{37}$$

where k is ~ 0.06 m$^{3/5}$s$^{-6/5}$ (Nienhuis et al., 2015) and $\theta(j) = (x_{s,j+1} - x_{s,j}) / \Delta y$. $\Psi$ is the angle dependence of the diffusivity (AM06), which we compute as follows:

$$\Psi(\phi_0 - \theta) = \cos^{1/5}(\phi_0 - \theta) \left[ \cos(\phi_0 - \theta)^2 - \tfrac{6}{5} \sin^2(\phi_0 - \theta) \right], \tag{38}$$

which we convolve with the normalized angular distribution of wave energy $E(\phi)$,

$$E(\phi_0) = \begin{cases} a \cdot h & \text{if} \quad -\tfrac{1}{2}\pi < \phi_0 < -\tfrac{1}{4}\pi \\ a + (1-h) & \text{if} \quad -\tfrac{1}{4}\pi < \phi_0 < 0 \\ (1-h)(1-a) & \text{if} \quad 0 < \phi_0 < \tfrac{1}{4}\pi \\ (1-a)h & \text{if} \quad \tfrac{1}{4}\pi < \phi_0 < \tfrac{1}{2}\pi \end{cases} , \tag{39}$$

to generate a long-term, wave-climate averaged shoreline diffusivity for every alongshore location j.



We rewrite the shoreline diffusion eq. (36) into,

$$\frac{x_{s\,j}^{\,n+1} - x_{s\,j}^{\,n}}{\Delta t} = \frac{D_j}{2} \frac{\left(x_{s\,j+1}^{\,n+1} - 2x_{s\,j}^{\,n+1} + x_{s\,j-1}^{\,n+1}\right) + \left(x_{s\,j+1}^{\,n} - 2x_{s\,j}^{\,n} + x_{s\,j-1}^{\,n}\right)}{\Delta y^2} + \frac{\Delta x_{s,j,ext}}{\Delta t},\tag{40}$$

where $n$ and $j$ denote the specific time and space locations. We solve this equation by inverting this nearly tri-diagonal matrix,

$$\begin{bmatrix} \ddots & \ddots & 0 & 0 & -\beta_j \\ \ddots & \ddots & \ddots & 0 & 0 \\ 0 & -\beta_j & 1+2\beta_j & -\beta_j & 0 \\ 0 & 0 & \ddots & \ddots & \ddots \\ -\beta_j & 0 & 0 & \ddots & \ddots \end{bmatrix} \begin{bmatrix} \vdots \\ x_{s\,j-1}^{\,n+1} \\ x_{s\,j}^{\,n+1} \\ x_{s\,j+1}^{\,n+1} \\ \vdots \end{bmatrix} = \begin{bmatrix} \vdots \\ \vdots \\ x_{s\,j}^{\,n} + \beta_j\left(x_{s\,j+1}^{\,n} - 2x_{s\,j}^{\,n} + x_{s\,j-1}^{\,n}\right) + \Delta x_{s,j,ext} \\ \vdots \\ \vdots \end{bmatrix},\tag{41}$$

where $\beta_j = D_j^n \Delta t / 2 / \Delta y^2$. Because we use $D$ at $n$ instead of $n+1$ this is simply a linear diffusion equation. Indices in the lower right and upper left corner indicate periodic boundary conditions. The source $x_{s,j,ext}$ (m) can be described by,

$$\Delta x_{s,j,ext} = \Delta x_{s,j,ow,sf} + \Delta x_{s,j,inlet},\tag{42}$$

representing offshore and onshore sediment fluxes that can erode and accrete the shoreline, and flood-tidal delta deposition that acts as a littoral sink.

The shoreline model is unconditionally stable and second-order accurate in space and time. We discretize the coastline into cells with width $\Delta y$ (typically 100 m). We use a timestep $\Delta t$ (typically 0.05 yr) to ensure smooth inlet migration and reasonably accurate shoreline change. However, we note that the overwash and inlet elements of this model are not solved by equation 41, and are therefore not second-order accurate nor unconditionally stable. Section 5 presents the grid and time resolution tests.

### 3.5    Other moving boundaries

At the end of each timestep, we update the shoreface toe position $x_t$,

$$x_t^{\,n+1} = x_t^{\,n} + \Delta x_{t,ow},\tag{43}$$

back-barrier location ($x_b$),

$$x_b^{\,n+1} = x_b^{\,n} + \Delta x_{b,ow} + \Delta x_{b,inlet} + \Delta x_{b,breach},\tag{44}$$

and barrier height ($H$),

$$H^{\,n+1} = H^{\,n} + \Delta H_{ow} + \Delta H_{inlet},\tag{45}$$

independently for all alongshore locations $j$, and run another time step.



### 3.6 Model output

After a model simulation (typically 10 ka) we obtain shoreline, backbarrier, and shoreface morphodynamics for different scenarios given by, for example, SLR rates, wave climates, and tidal conditions. One aspect of particular interest, and the primary motivation for this model, is the transgressive flux due to inlet activity. We define a ratio $F$,

$$F = \frac{Q_{inlet}}{Q_{overwash} + Q_{inlet}}, \qquad (46)$$

where $Q_{overwash} = \Delta y \cdot \sum_y Q_{ow,b}$ (m$^3$ yr$^{-1}$), and $Q_{inlet}$ is the along-coast average transgressive sediment flux by inlet formation

and flood-tidal delta deposition for all inlets,

$$Q_{inlet} = \frac{\Delta x_{b,breach}}{\Delta t} \frac{(H + D_{inlet})}{W_b} + \sum_{i=1..n_{inlet}} (1 - \beta - \beta_r) Q_{s,in,i} . \qquad (47)$$

$F$ quantifies the fraction of the total transgressive flux due to inlets and can range from 0 to 1.

### 3.7 Stratigraphy module

Aside from the usual output such as transgressive fluxes, inlet morphodynamics, and barrier island change, the model can also compute the synthetic stratigraphy of a barrier at a certain location $x_{strat}$ for all grid cells $j$ (Fig. 6). When $x_b$ exceeds $x_{strat}$, the model saves the location $j$, lagoon depth $D_{lagoon}$, the sediment deposit thickness (i.e. $D_{lagoon} - z$), and the responsible process, either flood-tidal delta deposition or storm overwash. While $x_s < x_{strat} < x_b$, we record the height of the barrier $H$ as dune construction or erosion bounded vertically by $z$ and $H$. If an inlet is present, it erodes the deposit up to a depth $d_{inlet}$. Inlet migration forms sedimentary facies between $d_{inlet}$ and $z$. If an inlet is closed, it forms inlet fill facies. These barrier island facies allow us to compare model output to geological reconstructions of barrier islands (e.g., Mallinson et al., 2010).

## 4 Example model runs

### 4.1 Model without inlets

We first investigated a simulation without inlets, focusing on the effect of alongshore transport gradients and barrier overwash on barrier evolution. As we might expect, in the case of no inlets and uniform initial conditions alongshore, the barrier retreats uniformly and alongshore sediment fluxes do not affect barrier response. We also performed a model experiment with an initially variable barrier width driven by spatial changes in the bay shoreline location (Fig. 5). In this scenario, the initially narrower barrier stretches overwash more than the wider stretches, and therefore transgress faster. As shoreline curvatures increases, the magnitude of the alongshore sediment fluxes directed to the narrow stretches also increases, which reduces the width of the initially wider stretches. Interestingly, we find that time lags in shoreline interconnectivity can cause the initially rapidly transgressing stretch stay in place and eventually become landward of other



portions of the coast, a phenomena also reported by Ashton & Lorenzo-Trueba (2018). Eventually, after a few oscillations that can last for hundreds of years, the barrier approaches a spatially uniform migration rate (Fig. 5).

## 4.2     Model with inlets

Including tidal inlets we see a richer set of model dynamics. In an example model simulation we investigated barrier change
for a SLR rate of 2 mm yr$^{-1}$, a wave height of 1 m, and a tidal range of 1 m. After an initial spin-up phase associated with large overwash fluxes, barrier island response stays highly dynamic and does not converge to an equilibrium response, despite imposing constant boundary conditions (Fig. 6). Inlets open, close, and interact, and migrate preferentially with the direction of the littoral drift. Inlet migration rates vary gradually, and inlet sediment distribution is initially dominated by alongshore sediment bypassing and gradually becomes more flood-tidal delta dominated (Fig. 6e). The inlet transgressive
sediment flux is highest when the flood-tidal delta deposition and alongshore sediment bypassing are roughly equal (Fig. 6f). Barrier stratigraphy at that time shows that inlet migration facies make up most of the barrier, even though not all of the transgression is due to the inlet (Fig. 6b).

## 5     Model tests

### 5.1     Conservation of mass

To investigate model mass conservation, we summed the volume of the barrier and offshore deposits (Fig. 7). Comparison to an identical model without inlets shows that slight losses and gains can be attributed to inlet morphodynamics, likely inlet migration and closure (Fig. 7b). For example, we do not track the sediment lost or gained as inlets change their cross-sectional area from an initial breach width. We also assume that increases in the back-barrier location can be considered small enough so that there is one depth $D_{lagoon}$, whereas in reality these deposits exist on a surface with slope $s_{background}$.
Regardless of these assumptions, model volume (offshore deposits and the barrier island itself) does not dependent on the time step $\Delta t$ and the grid length $\Delta y$; these values only deviate a few percent around their mean, with no obvious trend in time (Fig. 7b).

### 5.2     Comparison to the 1D model

For model verification, we compared model results to the original cross-shore model of barrier change that only includes
overwash (LTA14) (Fig. 8). Our model without inlets produces the same dynamics as the original cross-shore model (LTA14), resulting in the same overwash flux (Fig. 8a). Comparing the cross-shore model to the BRIE model with inlets (forced non-uniformity) we see some clear differences. Even though the average shoreline location along the 100km barrier follows roughly the same trajectory (Fig. 8c) and therefore has a similar transgression (erosion) rate (Fig. 8b), the individual locations vary significantly. The straight barrier reproduced by the BRIE model without inlets is now variable alongshore.



Transgression rates vary from -2 m/yr (progradation) to +10 m/yr (erosion). Even though the overall trajectory is a result from the sea level history and the passive inundation of the main (non-barrier) coast (Wolinsky and Murray, 2009) (Fig. 8c), significant deviations from this trend appear and are reflected in the overwash rates (Fig. 8a). In particular, the inlet transgressive sediment flux rates are variable.

5 ## 5.3  Sensitivity to grid resolution and timestep

We investigated the sensitivity of the model output ($Q_{overwash}$, $Q_{inlet}$, and $F$) by varying the grid resolution and time resolution, and holding all other parameters constant. In general, we find that these fluxes vary approximately ~20% between different settings (Fig. 9). These deviations appear only in simulations that include inlets, and are likely caused by a sensitivity to small perturbations such as random wave angles. For example, comparing multiple simulations with equal settings, including 10 grid and time, we obtain a variability in $F$ (Fig. 10), with a standard deviation of 0.025. Sensitivity to grid spacing and time steps can also be caused by discretization of inlet migration rates and distances (eq. 35)

## 6  Model evaluation

It is challenging to evaluate long-term barrier island models against natural examples. Given their erosional nature, long-term records or barrier dynamics are scarce (Mellett and Plater, 2018). Thus, instead of a direct comparison to natural 15 examples, we evaluate our model by exploring the sensitivity of the model output to a variety of boundary conditions (Fig. 11). We find that, even though individual simulations show great variability over time (Fig. 6), longer timescale dynamics of barrier islands present physically meaningful relationships with model boundary conditions. For example, wave height tends to increase the effect that inlets have on barrier transgression, likely by making inlets more wave-dominated and by increasing their migration rates. Inlets are most effective for intermediate back-barrier depths, whereas overwash volumes 20 are highest for deeper back-barrier depths. The greater effect of inlets for an intermediate depth could be because flood-tidal delta growth is enhanced, thereby restricting tidal flow and forcing the opening of inlets elsewhere (Fig. 11).

## 7  Discussion and conclusion

We have built a 2-D barrier island model (i.e., the BRIE model) to simulate barrier island response to SLR that couples alongshore sediment transport processes, storm overwash, and tidal inlet morphodynamics. The mathematics of the approach 25 are verified by comparing model predictions without inlets against the LTA14 cross-shore model. We also show that sediment volume is conserved with sufficient accuracy under a wide range of scenarios. Model results demonstrate that feedbacks between shoreface dynamics, barrier overwash, and alongshore transports processes can result in a complex history of interconnected behavior between the shoreline and barrier location. Moreover, we find that the relative importance of tidal inlets and storm overwash in transporting sediments onshore during barrier landward migration can significantly vary



as a function of a wide range of factors, including sea-level rise rate, wave climate, barrier and inlet geometries, and antecedent topography. Overall, model results highlight the importance of the interplay between cross-shore and alongshore processes, particularly tidal processes, in understanding future and past barrier response to sea-level rise.

The BRIE modeling framework does not aim at reproducing the evolution of any particular field location. Instead, we focus on exploring the relative role of tidal and overwash fluxes on the response of barriers to SLR, which requires omitting processes that could also play a significant role. For instance, the BRIE model does not account for human activities and coastal protection strategies along the coast (e.g., sea walls, groins, beach nourishment), which are known to affect coastal response at different spatial and temporal time scales (Brad Murray et al., 2013; Jin et al., 2013). Rather than accounting for

marsh-lagoon dynamics in the backbarrier environment, which can potentially influence the rate of barrier landward migration under sea-level rise (FitzGerald et al., 2008; Lorenzo-Trueba and Mariotti, 2017), we define a fine sediment thickness based on the lagoon depth and the basement slope. We also ignore the stochastic nature of storms, as well as the potential dynamic influence of shoreface lithology. Given its simplicity, however, the BRIE modeling framework can be extended to account for additional processes that might affect barrier evolution, including the ones mentioned above.

**Code availability**

The model is written in MATLAB. Source code and user manual are available at the CSDMS repository and at GitHub under an MIT license:

- csdms.colorado.edu/wiki/Model:Barrier_Inlet_Environment_(BRIE)_Model, doi:10.5281/zenodo.1218142
- https://github.com/csdms-contrib/Barrier_Inlet_Environment_BRIE_Model

**Data availability**

Model output used to generate figure 6 and supplemental animation S1 can be found in supplemental dataset S1.

**Video supplement**

Supplemental animation S1 describes the transgression of an example barrier island simulated using BRIE.

**Author contribution**

25   J.H.N. conceived the study and wrote the model. Both authors contributed to data analysis and manuscript preparation.





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





**Figures and Tables**

**Table 1. Model variables and their dimensions. Shortened references are: LTA14 (Lorenzo-Trueba & Ashton, 2014), B80 (Bowen, 1980), N15 (Nienhuis et al., 2015), R13 (Roos et al., 2013), SH85 (Sallenger & Holman, 1985), SZ09 (de Swart & Zimmerman, 2009), AM06 (Ashton & Murray, 2006), NA16 (Nienhuis & Ashton, 2016).**

| Name | Value | Units | Explanation |
|---|---|---|---|
| (independent variables) | | | |
| $\rho_w$ | 1025 | kg m$^{-3}$ | density of water |
| $R$ | 1.65 | - | submerged specific gravity of sediment |
| $g$ | 9.81 | m s$^{-2}$ | gravitational acceleration |
| $H_{crit}$ | 2 | m | critical barrier height (LTA14) |
| $T_p$ | 10 | s | peak wave period |
| $s_{background}$ | varied | - | background slope (LTA14) |
| $d_{50}$ | $1 \cdot 10^{-4}$ | m | median grain size |
| $e_s$ | 0.01 | - | suspended sediment transport efficiency factor (B80) |
| $c_s$ | 0.01 | - | friction factor (B80) |
| $k$ | 0.06 | m$^{3/5}$ s$^{-6/5}$ | alongshore sediment transport constant (NAG15) |
| $T_{storm}$ | varied | yr | minimum period between inlet forming storms |
| $L_{min}$ | varied | m | minimum distance between tidal inlets (R13) |
| $\omega$ | $1.4 \cdot 10^{-4}$ | s$^{-1}$ | offshore tidal radial frequency |
| $\gamma_{aspect}$ | 0.01 | - | inlet aspect ratio (depth / width) |
| $n$ | 0.05 | s m$^{-1/3}$ | manning roughness coefficient (vegetated tidal lagoon) |
| $\gamma$ | 0.4 | - | wave breaking criterion (SH85) |
| $u_{eq}$ | 1 | m s$^{-1}$ | tidal inlet equilibrium velocity (SZ09) |
| $W_{b,crit}$ | varied | m | critical barrier width (LTA14) |
| $Q_{ow,max}$ | varied | m$^3$ m$^{-1}$ yr$^{-1}$ | maximum barrier overwash flux (LTA14) |
| $D_{lagoon}$ | varied | m | back barrier lagoon depth |
| $H_s$ | varied | m | deep-water significant wave height |
| a | varied | - | fraction of waves approaching from the left, looking offshore (AM06) |
| h | 0.2 | - | fraction of waves approaching at a high angle (> 45˚) (AM06) |

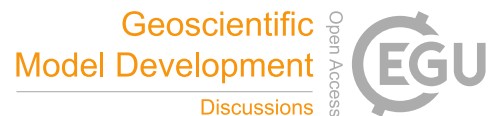



| Name | Value | Units | Explanation |
|------|-------|-------|-------------|
| $\dot{z}$ | varied | m yr$^{-1}$ | Sea-level rise rate |
| $\Delta y$ | varied | m | alongshore grid spacing |
| $\Delta t$ | varied | yr | time step |
| $a_0$ | varied | m | offshore tidal amplitude |
| $f_{marsh}$ | varied | - | fraction of the lagoon surface area not contributing to the tidal prism |
| (dependent variables) | | | |
| $s_{sf}$ | $D_t / (x_s - x_t)$ | - | shoreface slope (LTA14) |
| $s_{sf,eq}$ | eq. 15 | - | equilibrium shoreface slope |
| $D_T$ | eq. 14 | m | shoreface depth (LTA14) |
| $D$ | eq. 37 | m$^2$s$^{-1}$ | shoreline diffusivity (AM06) |
| $f$ | eq. 11 | - | fraction fines in the barrier |
| $x_t$ | eq. 1 | m | position of the shoreface toe (LTA14) |
| $x_s$ | eq. 2, 35, 36 | m | position of the shoreline (LTA14) |
| $x_b$ | eq. 3, 17, 34 | m | position of the back barrier (LTA14) |
| $H$ | eq. 4 | m | height of the barrier (LTA14) |
| $Q_{ow,b}$ | eq. 6 | m$^3$ m$^{-1}$ yr$^{-1}$ | overwash flux deposited in the back barrier (LTA14) |
| $Q_{ow,h}$ | eq. 5 | m$^3$ m$^{-1}$ yr$^{-1}$ | overwash flux deposited on top of the existing barrier (LTA14) |
| $V_{d,h}$ | eq. 8 | m$^3$ m$^{-1}$ | barrier height deficit (LTA14) |
| $V_{d,w}$ | eq. 7 | m$^3$ m$^{-1}$ | barrier width deficit (LTA14) |
| $z_0$ | $H_s/\gamma$ | m | minimum integration depth for shoreface flux |
| $k_{sf}$ | eq. 12 | m$^3$ m$^{-1}$ yr$^{-1}$ | shoreface response rate  (LTA14) |
| $w_s$ | eq. 16 | m s$^{-1}$ | settling velocity |
| $Q_{sf}$ | eq. 9 | m$^3$ m$^{-1}$ yr$^{-1}$ | shoreface flux |
| $W_b$ | $x_b - x_s$ | m | barrier width |
| $\phi_0$ | eq. 24 | - | offshore wave direction |
| $\theta$ | arctan($dx_s/dy$) | - | shoreline angle |
| $Q_{s,in}$ | eq. 23 | m$^3$ m$^{-1}$ yr$^{-1}$ | alongshore sediment flux into inlet |





| Name | Value | Units | Explanation |
|---|---|---|---|
| $D_{inlet}$ | $(A_{inlet}\gamma_{aspect})^{1/2}$ | m | inlet depth |
| $W_{inlet}$ | $(A_{inlet}/\gamma_{aspect})^{1/2}$ | m | inlet width |
| $W_{lagoon}$ | $z/s_{lagoon} - x_b$ | m | cross-shore width of the lagoon |
| $L_{lagoon}$ | | m | alongshore length of the lagoon draining to a particular tidal inlet |
| $A_{inlet}$ | eq. 21 | m$^2$ | inlet cross-sectional area |
| $u$ | eq. 22 | m s$^{-1}$ | inlet flow velocity |
| $A_b$ | $W_b (H + D_{inlet})$ | m$^2$ | barrier cross-sectional area |
| $A_{b,downdrift}$ | $W_b (H + D_{inlet})$ | m$^2$ | barrier cross-sectional area downdrift of an inlet |
| $A_{b,updrift}$ | $W_b (H + D_{inlet})$ | m$^2$ | barrier cross-sectional area updrift of an inlet |
| $\alpha$ | eq. 26 | - | fraction of $Q_{s,in}$ deposited as new barrier (NA16) |
| $\beta$ | eq. 28 | - | fraction of $Q_{s,in}$ that bypasses the inlet (NA16) |
| $\delta$ | eq. 30 | - | fraction of $Q_{s,in}$ that deposits as flood-tidal delta (NA16) |
| $\alpha_r$ | eq. 27 | - | eroded barrier deposited as new barrier, fraction of $Q_{s,in}$ (NA16) |
| $\beta_r$ | eq. 29 | - | eroded barrier deposited in the littoral zone, fraction of $Q_{s,in}$ (NA16) |
| $\delta_r$ | eq. 31 | - | eroded barrier deposited in flood-tidal delta, fraction of $Q_{s,in}$ (NA16) |
| (model output) | | | |
| $Q_{overwash}$ | | m$^3$m$^{-1}$yr$^{-1}$ | Transgressive flux due to storm overwash |
| $Q_{inlet}$ | eq. 47 | m$^3$m$^{-1}$yr$^{-1}$ | Transgressive flux due to tidal inlets |
| $F$ | eq. 46 | - | Fraction of the transgressive flux from inlets |





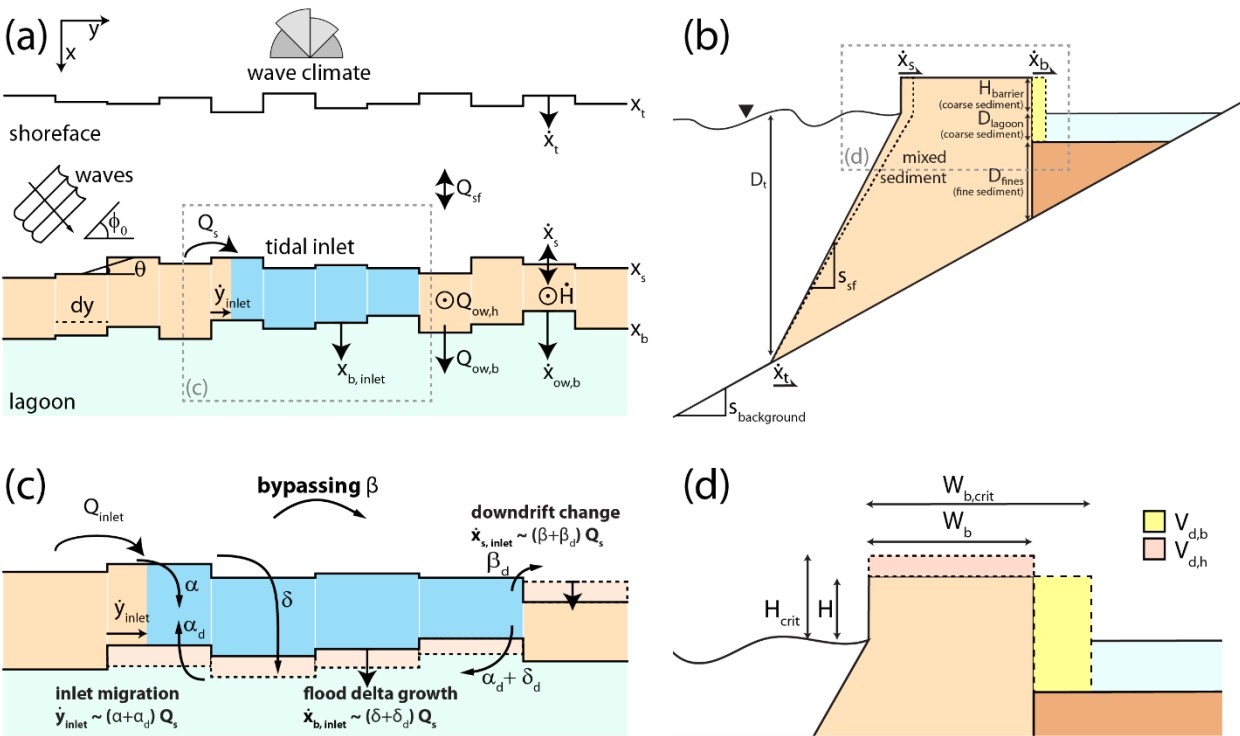

**Figure 1.** Schematized model domain in (a) the plan-view, highlighting the three moving boundaries, the shoreface toe, shoreline, and the back-barrier (lagoon) location, and the sediment fluxes that determine their coupling. Vectors indicate the direction of potential changes, with the dot symbolizing a movement up. (b) Implementation of fine sediment dynamics into the barrier overwash model. (c) a close-up of (a), showing the littoral sediment fractionation within an inlet and its translation to barrier change. (d) A close-up of (b), showing the barrier volume deficit approach. See table (1) for model variable names and units.

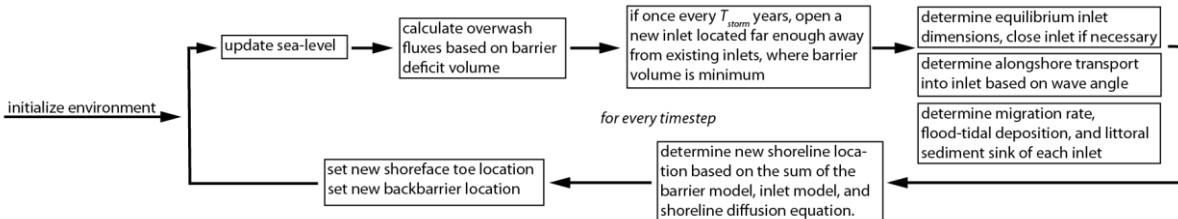

**Figure 2.** Model structure, showing the time loop in which the model updates SLR, and calculates resulting barrier island change.





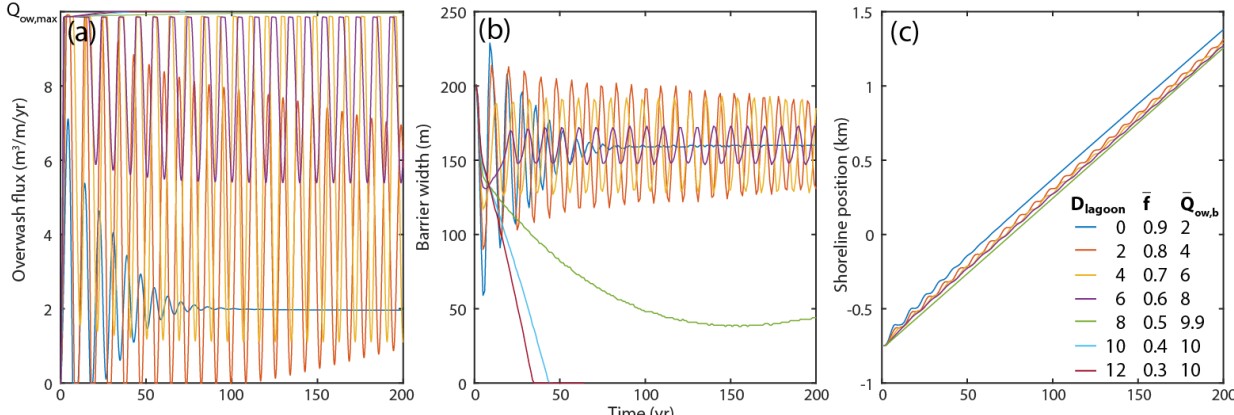

**Figure 3. Effect of lagoon depth (fine sediment fraction) on (a) the overwash flux, (b) the barrier width, and (c) the shoreline location. More overwash flux is needed to maintain a barrier with a deep lagoon, resulting in barrier drowning for a lagoon depth of > 8 m, where the required barrier overwash flux is greater than the maximum**
5 **potential overwash flux $Q_{ow,max}$.**



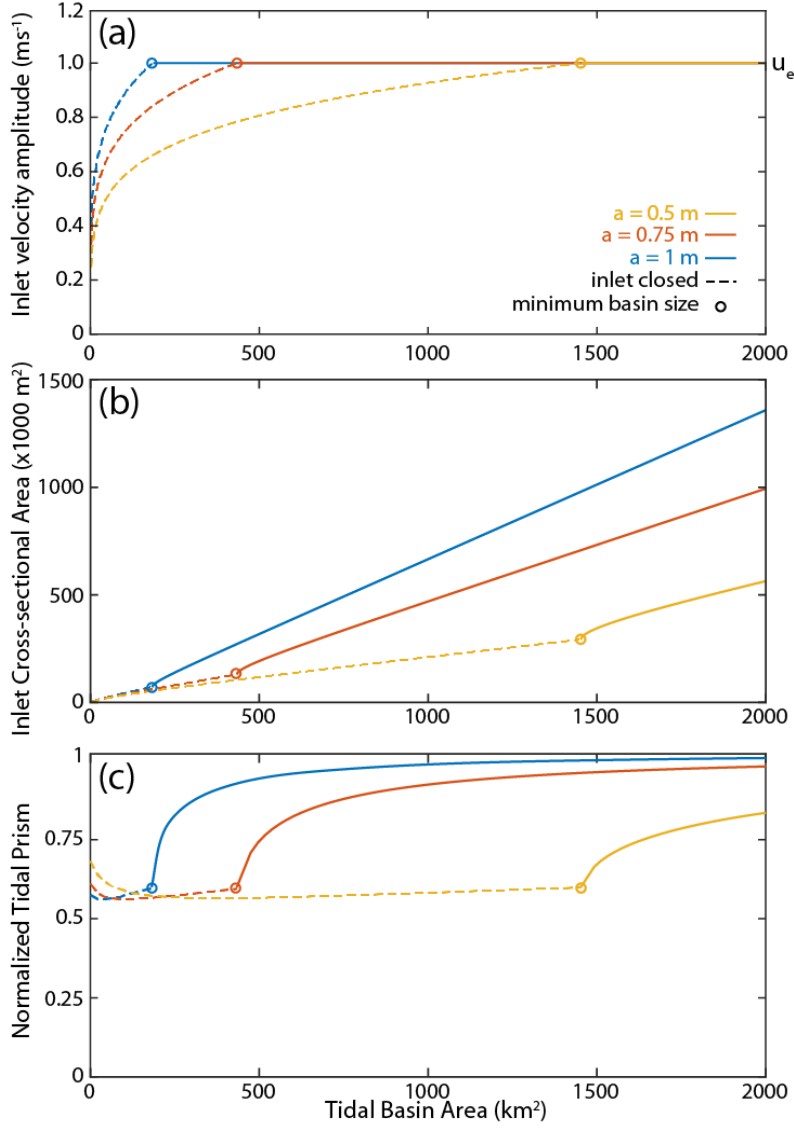

**Figure 4. Analytical solutions to the equations that govern tidal inlet size (eq. 21) as a function of tidal basin area and offshore tidal amplitude, illustrating that (a) inlets close if a velocity amplitude of 1 m s-1 cannot be maintained, (b) inlet cross-sectional area is dependent on tidal amplitude and tidal basin area, and (c) that lagoon and inlet friction reduce the volume of water transported through the inlet, which is less than the potential tidal prism (offshore tidal range multiplied by intertidal area).**

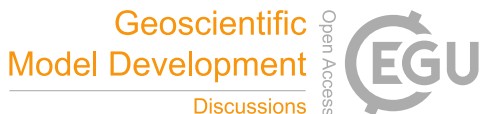



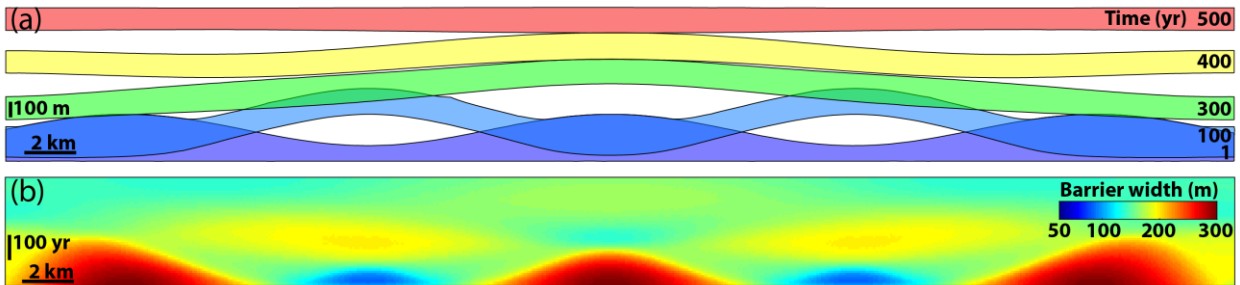

**Figure 5. BRIE without inlets showing barrier response to variations in the initial back-barrier position xb (see also Ashton & Lorenzo-Trueba, 2018), (a) $x_s$ and $x_b$ at 5 instances, (b) barrier width as a function of time. Minima in barrier width drive faster transgression, which in turn results in wider barriers through accumulation of alongshore sediment.**

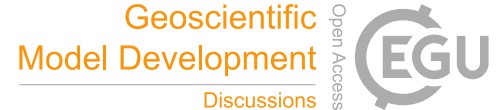



**Figure 6.** Example model run showing (a) a barrier island including tidal inlet through time, (b) inlet facies after approximately 1300 years, (c) inlet location through time (such that the slope of the line represents the migration rate), (d) average inlet migration rate, (e) the average fraction of alongshore sediment brought into the inlet transported to the downdrift coast ($\beta$), the flood-tidal delta ($\delta$), and the barrier itself ($\alpha$), (f) the transgressive sediment flux due to overwash and due to flood-tidal delta deposition.

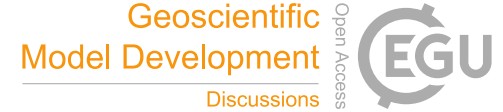

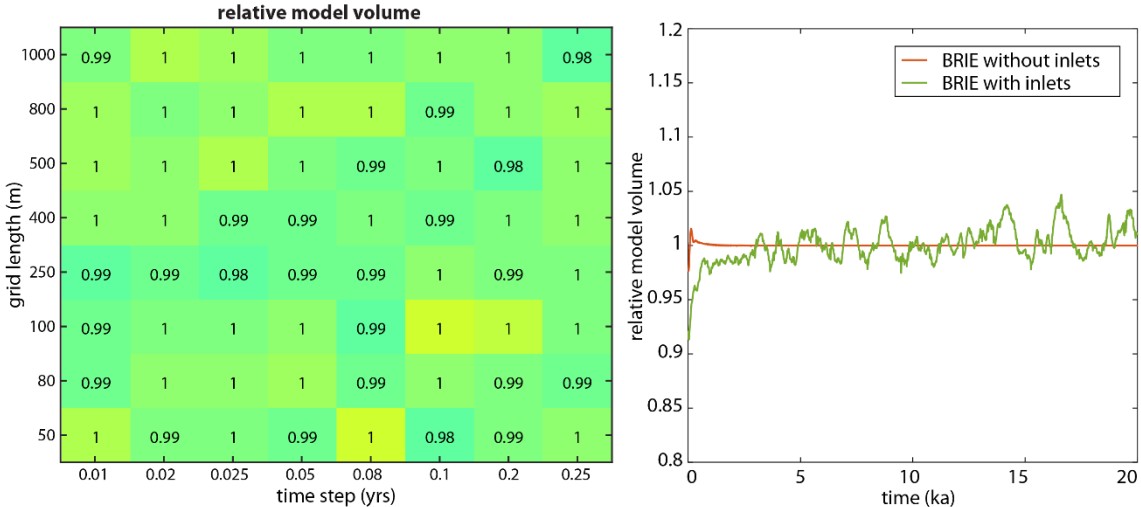

**Figure 7. Model volume (barrier volume and offshore deposits) relative to the analytically determined model volume for (a) averaged for different time steps and alongshore grid lengths, and (b) through time, showing mass conservation.**

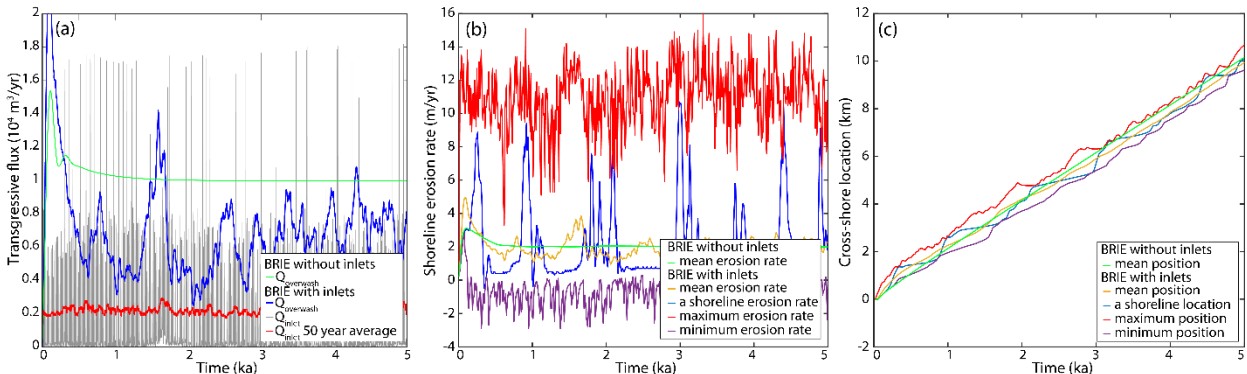

**Figure 8. Comparison of (a) transgressive fluxes, (b) shoreline erosion rates, and (c) shoreline locations for the BRIE model without inlets to the model with inlets. Inlets induce stochasticity resulting from inlet opening, growing, migrating, and closing.**





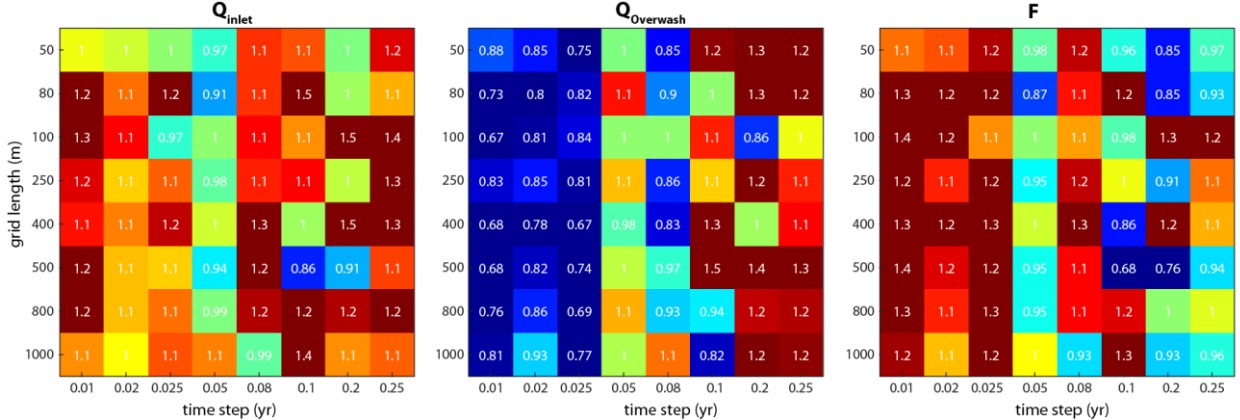

**Figure 9. Tidal induced transgressive sediment flux, storm overwash induced transgressive sediment flux, and the ratio $F$ as a function of grid length and time step normalized by their values at a timestep of 0.05 years and 100 m alongshore grid discretization.**

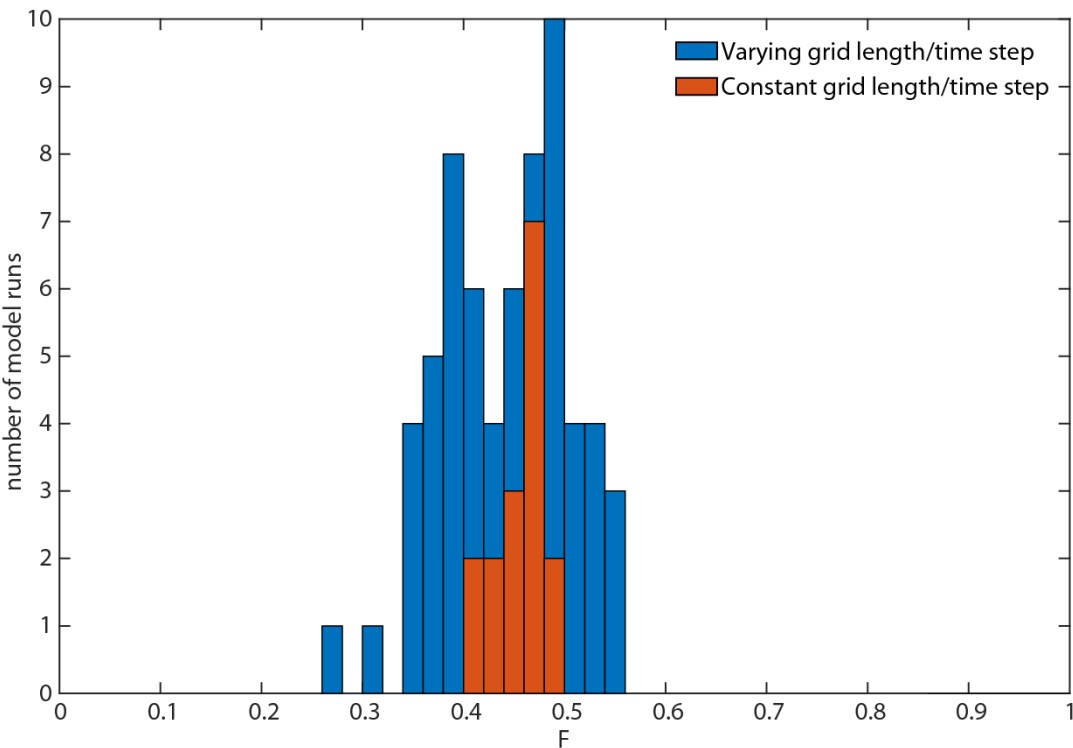

**Figure 10. Variability in the inlet fraction of the transgressive sediment flux $F$, for varying grid lengths $\Delta y$ and time steps $\Delta t$ and for constant $\Delta y$ and $\Delta t$, highlighting the fraction of the variability in $F$ due to inherent model variability.**



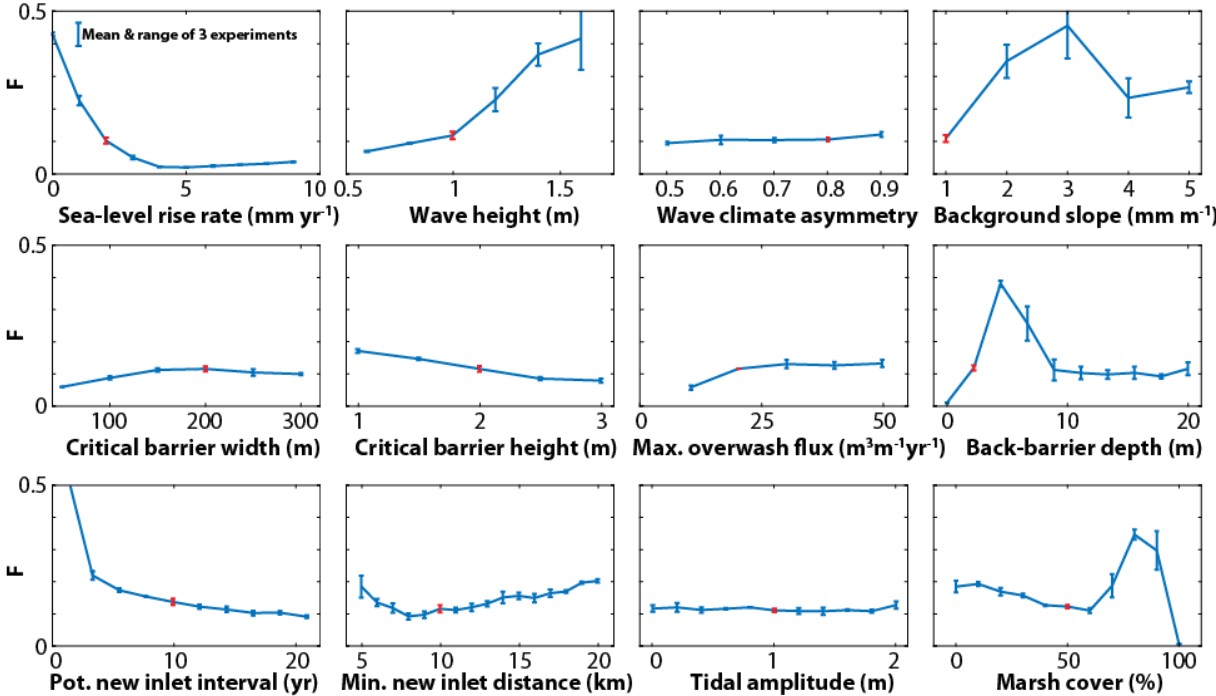

**Figure 11. Fraction F for varying model parameters, showing the mean and the range obtained for three different experiments. In red the value of the specific parameter that is held constant across the other simulations.**