# Peer review of "Simulating barrier island response to sea-level rise with the barrier island and inlet environment (BRIE) model v1.0"

_Geoscientific Model Development, 2019_

## Referee Comment (RC1) · Eli D Lazarus (Referee) · 30 Mar 2019

This manuscript describes a numerical model of long-term barrier island evolution that includes the dynamical effects of inlets.

First, I commend the authors on a clear and cogent submission. As presented, the work serves the two functions it needs to address: the technical detail required by model users who may find themselves deep in the numerical machinery; and the explanatory summary required by readers looking for a sense of what the model does, and how.

My remarks have mostly to do with framing. At P1/L24–26, the authors state, "...there

exists a critical gap on understanding how barriers respond to change generally, and [to] sea-level rise specifically." This premise extends into the first two sections (Introduction and Background – and it appears in the Abstract). I flag it here because I don't think the statement is accurate – and the Background subsections essentially demonstrate its inaccuracy. (In the interest of full disclosure, I've been called out before for making a similar claim. The person was right to make the point – and it's equally valuable here.)

If current coastal science understands anything about barrier dynamics, it seems to me it's how they "respond to change generally" and to "sea-level rise specifically." I'm not sure what the authors mean by "respond to change generally" – but regardless, the "critical gap," as defined, isn't really what this model is ultimately concerned with.

All that is to say: a more precise statement of the "critical gap" early in the Introduction would go a long way toward streamlining the manuscript. The authors need look no further than their Background section, which is precise. The authors move rapidly and confidently through major developments in the discipline – and in doing so, they make their case for how their BRIE model represents an advance.

To me, the critical gap that BRIE addresses is how barriers, as they transgress landward with sea-level rise, ALSO adjust to alongshore sediment flux, inlet dynamics (initiation, migration, capture, erasure), and back-barrier sedimentation – and, in turn, how those adjustments affect barrier response to sea-level rise. (There are other gaps that this model does not address, such as the role of ecological feedbacks in barrier dunes and tidal wetlands, which must also inform evolution dynamics in fundamental ways. I do not mean to suggest that BRIE must model everything.) So, if the authors were to open the article with a clear paragraph along those lines (and propagate that framing through the rest of the first two sections), they would both nod to past contributions and chalk out the space in which they are working.

In a similar vein, on P3L11-12, referring to overwash, the authors state that "longterm landward sediment flux is generally poorly constrained, and its relationship to modern overwash fluxes largely unexplored." Again, I would suggest that "unexplored" is perhaps a stronger statement than the authors mean to make? The relationship to modern overwash fluxes might remain unclear, but that doesn't make them unexplored. A comb through the document with fresh eyes will I hope reveal to the authors other such moments – they are subtle, but fixing them will avoid overstatement.

And a couple of very minor notes:

Save "LTA14" for mentions of that model, specifically? When referring to insights from that paper, I would cite the paper with its full in-text citation.

ASMITA needs some kind of introduction: "Coming from a different angle, the model known as ASMITA..." – but even then, I'm left wondering what the acronym stands for. A description does come in the sentence that follows, but it's a small step too late.

Again, hats off to the authors – I look forward to seeing this article in its final form.

СЗ

---

## Referee Comment (RC2) · Anonymous Referee #2 · 10 Jun 2019

This manuscript describes a modified version of the Lorenzo-Trueba and Ashton, 2014 numerical model. The authors have included many important additions such as accounting for fine sediment and tidal inlet dynamics. The background section does a good job describing previous studies and their limitation and how the BRIE model addresses many of these gaps, specifically the inclusion of tidal inlet dynamics which as the authors point out can contribute large volumes of sediment to the flood tidal delta. The authors present the numerical model development through detailed description of the processes included, equations used and assumptions made. While a table of variable is included, it would be helpful to include some of the more frequently used variable in the text after they are first presented.

The long term simulations presented do an efficient job indicating the stability of the BRIE model and the differences in the barrier island response due to the inclusion of inlet dynamics.

Some minor edits and notes: P07L22: Brenner et al. 2015 is not in references P15L27: ...transgressing stretch stay in... -> ...transgressing stretch to stay in... P16L20: ...does not dependent... -> ...does not depend... P17L27: ...alongshore transports processes... -> ...alongshore transport processes...

Fig2 b : increase the text size on some of the labels (e.g. coarse sediment and subscript) Fig7 : missing a,b labels that are described in the caption

---

## Author Comment (AC1) · 18 Jul 2019

*(author responses are italicized; line numbers refer to the track-changes document)*

**Reviewer #1**

This manuscript describes a numerical model of long-term barrier island evolution that includes the dynamical effects of inlets.

First, I commend the authors on a clear and cogent submission. As presented, the work serves the two functions it needs to address: the technical detail required by model users who may find themselves deep in the numerical machinery; and the explanatory summary required by readers looking for a sense of what the model does, and how.

*We thank Eli Lazarus for his helpful review.*

My remarks have mostly to do with framing. At P1/L24–26, the authors state, ". . .there exists a critical gap on understanding how barriers respond to change generally, and [to] sea- level rise specifically." This premise extends into the first two sections (Introduction and Background –

and it appears in the Abstract). I flag it here because I don't think the statement is accurate – and the Background subsections essentially demonstrate its inaccuracy. (In the interest of full disclosure, I've been called out before for making a similar claim. The person was right to make the point – and it's equally valuable here.)

If current coastal science understands anything about barrier dynamics, it seems to me it's how they "respond to change generally" and to "sea-level rise specifically." I'm not sure what the authors mean by "respond to change generally" – but regardless, the "critical gap," as defined, isn't really what this model is ultimately concerned with.

All that is to say: a more precise statement of the "critical gap" early in the Introduction would go a long way toward streamlining the manuscript. The authors need look no further than their

Background section, which is precise. The authors move rapidly and confidently through major developments in the discipline – and in doing so, they make their case for how their BRIE model represents an advance.

To me, the critical gap that BRIE addresses is how barriers, as they transgress landward with sea- level rise, ALSO adjust to alongshore sediment flux, inlet dynamics (initiation, migration, capture, erasure), and back-barrier sedimentation – and, in turn, how those adjustments affect barrier response to sea-level rise. (There are other gaps that this model does not address, such as the role of ecological feedbacks in barrier dunes and tidal wetlands, which must also inform evolution dynamics in fundamental ways. I do not mean to suggest that BRIE must model everything.) So, if the authors were to open the article with a clear paragraph along those lines (and propagate that framing through the rest of the first two sections), they would both nod to past contributions and chalk out the space in which they are working.

*We fully agree that the knowledge gap we described was somewhat vague. BRIE indeed*

*aims to quantify potential transgressive tidal fluxes for different coastal settings and sea-*

*level rise rates. There are likely to be feedbacks between these tidal fluxes and overwash*

*deposition, hence the need for a comprehensive model that includes both mechanisms for*

*barrier transgression.*

*That said, however, modern coastal science theories on long-term barrier dynamics are*

*not fully field-validated. The parameterizations for transgressive sediment fluxes, in*

*particular in relation to sea-level rise rates and wave climates, have not been thoroughly*

*tested. We agree that BRIE does not directly solve this problem and that, therefore, the*

*knowledge gap was not presented properly.*

*We have changed our knowledge gap in the abstract (L10) to: "Despite their socio-*

*economic and ecological importance, their future morphodynamic response to sea-level*

*rise or other hazards is poorly understood."*

*This change, we believe, better highlights the fact that our models ability to quantify*

*long-term transgressive fluxes and future barrier change are still very uncertain.*

*We now write in the introduction (L24):*

*"Despite their importance, there exists a critical gap in our ability to predict how*

*barriers will respond to coastal change generally, and sea-level rise (SLR) specifically. A*

*necessary condition for barrier islands to migrate landwards and keep up with SLR is*

*sufficient sediment transport from the barrier front to the top and back via overwash fan*

*deposition and flood-tidal delta formation (Armon and McCann, 1979; Inman and Dolan,*

*1989; Kraft, 1971; Lorenzo-Trueba and Ashton, 2014; Mallinson et al., 2010; Moore et*

*al., 2010). There is little information, however, regarding the potential magnitudes of*

*these landward sediment fluxes, and how these fluxes vary as a function of the coastal*

*setting, wave climate, or SLR. Recent models (e.g., Lorenzo-Trueba and Ashton, 2014)*

*have suggested formulations for overwash fluxes, but the potential role of tidal fluxes,*

*their feedbacks with overwash deposition, and the resulting ability of barriers to keep*

*pace with SLR, remains unclear."*

In a similar vein, on P3L11–12, referring to overwash, the authors state that "long term landward sediment flux is generally poorly constrained, and its relationship to modern overwash fluxes largely unexplored." Again, I would suggest that "unexplored" is perhaps a stronger statement than the authors mean to make? The relationship to modern overwash fluxes might remain unclear, but that doesn't make them unexplored. A comb through the document with fresh eyes will I hope reveal to the authors other such moments – they are subtle, but fixing them will avoid overstatement.

*We fully agree that "unexplored" is too strong a statement. We changed unexplored to*

*"not straightforward".*

And a couple of very minor notes: Save "LTA14" for mentions of that model, specifically?

When referring to insights from that paper, I would cite the paper with its full in-text citation.

*Adjusted*

ASMITA needs some kind of introduction: "Coming from a different angle, the model known as

ASMITA. . ." – but even then, I'm left wondering what the acronym stands for. A description does come in the sentence that follows, but it's a small step too late.

*Adjusted*

Again, hats off to the authors – I look forward to seeing this article in its final form.

**Reviewer #2**

This manuscript describes a modified version of the Lorenzo-Trueba and Ashton, 2014

numerical model. The authors have included many important additions such as accounting for fine sediment and tidal inlet dynamics. The background section does a good job describing previous studies and their limitation and how the BRIE model addresses many of these gaps, specifically the inclusion of tidal inlet dynamics which as the authors point out can contribute large volumes of sediment to the flood tidal delta. The authors present the numerical model development through detailed description of the processes included, equations used and assumptions made. While a table of variable is included, it would be helpful to include some of the more frequently used variable in the text after they are first presented.

*We thank the reviewer for their helpful review. We now present some of the more*

*important variables within the main text.*

The long term simulations presented do an efficient job indicating the stability of the BRIE

model and the differences in the barrier island response due to the inclusion of inlet dynamics.

Some minor edits and notes:

P07L22: Brenner et al. 2015 is not in references

*fixed*

P15L27: ...transgressing stretch stay in... –> ...transgressing stretch to stay in...

*fixed*

P16L20: ...does not dependent... –> ...does not depend...

*fixed*

P17L27: ...alongshore transports processes... –> ...alongshore transport processes...

*fixed*

Fig2 b : increase the text size on some of the labels (e.g. coarse sediment and subscript)

*fixed*

Fig7 : missing a,b labels that are described in the caption

*fixed*